# Surgical Treatment of Sialolithiasis Leads to Improvement in the Complete Blood Count

**DOI:** 10.3390/biology10050414

**Published:** 2021-05-07

**Authors:** Gal Avishai, Idan Rabinovich, Hanna Gilat, Gavriel Chaushu, Liat Chaushu

**Affiliations:** 1Department of Oral and Maxillofacial Surgery, Rabin Medical Center, Petach-Tikva 49414, Israel; GavrielCe@clalit.org.il; 2Department of Oral and Maxillofacial Surgery, The Maurice and Gabriela Goldschleger School of Dental Medicine, Tel-Aviv University, Tel Aviv 69978, Israel; idan2@mail.tau.ac.il; 3Department of Otolaryngology-Head and Neck Surgery, Rabin Medical Center, Petach Tikva 49414, Israel; hannagi2@clalit.org.il; 4Department of Periodontology and Implant Dentistry, The Maurice and Gabriela Goldschleger School of Dental Medicine, Tel-Aviv University, Tel Aviv 69978, Israel; liat.chaushu@gmail.com

**Keywords:** sialolithiasis, sialadenitis, anemia of chronic inflammation

## Abstract

**Simple Summary:**

Sialolithiasis is a disease in which inflammation and infection are caused in a salivary gland and its duct secretion system due to a formation of a sialolith (salivary stone) in the gland. Anemia of inflammation is a well described pathology where chronic inflammation causes a reduction in the red blood cell count is. In this study, we examined the complete blood count results of patients who underwent surgical removal of a sialolith and found that removal of the stone and cessation of the symptoms lead to an improvement in the complete blood count results. We believe that the improvement in blood count values after surgery is due to resolution of the anemia of inflammation. To our knowledge, this is the first report about the relationship between surgery for removal of a salivary stone and improvement in the wellbeing of patient expressed by blood count values.

**Abstract:**

Sialolithiasis is a chronic disease in which a sialolith (salivary stone) causes recurrent inflammation of the affected salivary gland. Anemia of inflammation is a well-described pathology in which a chronic inflammatory disease leads to a reduction in the red blood cell count, hemoglobin and hematocrit values. In this retrospective cohort study, we aim to find whether removal of the sialolith and alleviation of the inflammation affect the complete blood count results. We examined data regarding forty-nine patients who underwent surgery for the removal of a submandibular gland sialolith using the duct-stretching technique. Complete blood counts two years before and after the surgical procedure were collected. The average pre-procedure and post-procedure values were calculated for each patient to establish the average blood profile. The pre- and post-procedure values were compared to evaluate the effect of the surgical treatment on the blood profile. We found that the average blood count values for patients with sialolithiasis were towards the lower end of the normal range. Post-surgery, a significant increase in hematocrit, hemoglobin and red blood cell count was observed, which was more pronounced in the older age group and in patients with co-morbidities. We conclude that sialolith removal surgery is associated with significant improvement in the complete blood count values, especially in the elderly and in patients and with co-morbidities. The speculated pathogenesis is relative anemia of inflammation.

## 1. Introduction

Sialolithiasis is the formation of calcified stones within a salivary duct resulting in a mechanical obstruction to salivary gland secretion [1,2,3]. Clinical presentation may vary. Most commonly, cyclical sudden postprandial swelling of the effected salivary gland (‘mealtime syndrome’) with signs of sialadenitis due to retrograde infections from the oral cavity [1]. Sialolithiasis comprises 30% of salivary gland pathologies and may present in up to 1% of the general population [2,3].

The pathophysiological mechanism resulting in sialolith formation is not well understood [1,2,3]. Etiological factors believed to cause salivary stone formation are divided into a few major groups: Anatomical variability, salivary composition factors and hyposalivation. It has recently been suggested that different etiologies, namely drug-induced sialolithiasis due to hyposalivation versus other etiologies of sialolithiasis have unique characteristics regarding presentation and treatment outcome [4].

A wide range of treatments is available for sialolithiasis, dependent on the size and location of the sialolith. At the primary level of care, empiric antibiotic treatment and sialagogues to stimulate salivary secretion are used to treat acute infection and pain. Surgical interventions vary from minimally invasive techniques, such as sialendoscopy and sialolith extraction with preservation of the gland, to sialoadenectomy [1].

Previous studies have shown a direct link between oral diseases and systemic disease. The primary example of this correlation is periodontal disease and certain other systemic conditions, including: atherosclerotic vascular disease, pulmonary disease, diabetes, pregnancy-related complications, osteoporosis, neurodegenerative diseases and kidney disease [5]. The suspected pathways of the correlation between oral infection and systemic disease can be divided into two pathways: the first is a direct infection of bacteria from the oral flora to distant organs, either by hematogenous spread (e.g., endocarditis) or by seeding of the airway (lung abscess and pneumonia). The second pathway is by bacteremia or cytokinemia from an oral source, triggering inflammatory and/or immunological responses that result in damage to distant tissues or organ systems (e.g., atherosclerosis) [6]. Diabetes, more than any other systemic condition, demonstrates a distinctive bidirectional relationship with periodontal disease, and there is strong evidence that treating one condition positively impacts the other [7].

An emerging area of interest in relation to the implications of chronic disease with an inflammatory nature is the entity of Anemia of Inflammation (AI) [8,9]. This entity which is also referred to as anemia of chronic disease was described over 60 years ago as a mild to moderately severe anemia with hemoglobin levels from 7 to 12 g/dL. It has been theorized that the systemic inflammation causes decreased production of erythrocytes, accompanied by a modest reduction in erythrocyte survival [8]. AI, similar to iron-deficiency anemia, is characterized by low serum iron levels (hypoferremia), but it differs from iron-deficiency anemia in that iron stores are preserved in marrow macrophages, as well as in splenic and hepatic macrophages, which recycle senescent erythrocytes. Whereas erythrocytes in iron-deficiency anemia are often small (low mean corpuscular volume) and hemoglobin-deficient (low mean corpuscular hemoglobin concentration), the erythrocytes in anemia of inflammation most often appear normal. The pathogenesis of AI has been described as a reflection of a physiological immune driven response reducing iron levels in a host defense mechanism. This reduction of iron levels deprives pathogens from accessible iron, which is an important nutrient for their proliferation [9].

In older patients, AI is associated with decreased survival and impaired health-related quality of life [10]. However, it is not understood whether AI is a predictor of adverse outcomes of chronic disease in the aging population or an etiological factor involved in aggravating the underlying chronic condition [11]. In patients of all ages with specific diseases, improvement in anemic state has been shown to have a beneficial impact on morbidity and mortality [12]. Current treatment approaches for AI are directed at treating the underlying disease and are the most likely to provide a meaningful outcome in patients [9].

The importance of inflammation and consequently the effect on anemic state in a major systemic inflammatory state such as rheumatoid arthritis (the prototypical disease model for AI) are well-established in the literature [13]. Findings of a more limited inflammatory process and relative AI were demonstrated in a variety of periodontal conditions [14]. A meta-analysis composed of nine case-control studies found a decrease in hematological parameters in chronic periodontitis compared to healthy controls [14]. An interventional study on generalized chronic periodontitis patients also demonstrated an improvement in the state of anemia [15]. These findings provide evidence that chronic bacterial infections of the head and neck, leading to inflammatory response, affect the systemic anemic state, similar to other chronic conditions.

Since sialolithiasis is a chronic bacterial infectious disease causing a generalized inflammatory response, it may be hypothesized that it could lead to AI. The aim of the current study was to assess whether sialolithiasis causes AI and evaluate the anemic status after a surgical removal of a submandibular gland sialolith using the duct-stretching technique [16]. The null hypothesis of the present study was that sialolithiasis and surgical treatment for removal of the sialolith are not associated with alterations in complete blood count (CBC).

## 2. Materials and Methods

### 2.1. Patient Group

Retrospective cohort study of patients who underwent surgical removal of a submandibular sialolith using the duct-stretching technique at the Rabin Medical Center, Petach Tikva, Israel between 2013 and 2019. The inclusion criteria were: availability of pre-operative history and post-operative follow-up for at least 3 months, as well as availability of pre-operative and post-operative CBC. Exclusion criterion was: anemia due to known causes (aplastic anemia, sickle cell anemia, rheumatoid arthritis, leukemia, iron deficiency anemia, vitamin deficiencies). All patients received detailed information regarding the procedure and gave their full consent. Surgical intervention was similar for all patients and comprised of intra-oral access duct-stretching surgery for sialolith removal from the submandibular salivary gland, with preservation of the gland [16].

### 2.2. Blood Test Collection Protocol

For each patient, CBC results in a span of 2 years prior to and 2 years following the sialolith removal procedure were collected from patients’ medical history records. CBC parameters collected included: WBC (white blood count), RBC (red blood count), HGB (hemoglobin), HCT (hematocrit), MCH (mean corpuscular hemoglobin), MCHC (mean corpuscular hemoglobin concentration), MCV (mean corpuscular volume) and PLT (platelet count). In cases of multiple procedures performed (e.g., endoscopy followed by surgery), blood tests were collected in a span of 2 years previous to the first procedure and 2 years following the last procedure.

The initial blood profile data collection was based on all blood tests performed, pre- and post-procedure. In order to generate a value representing the base (“chronic”) pre- and post-procedure status for each patient, and to eliminate confounding values, an exclusion regime was adopted. The exclusion criteria for blood samples consisted of: tests referred from emergency room, inpatient hospitalizations, hospitalization immediately and one month following sialolith removal surgery and diagnostic blood tests in acute conditions (e.g., infection, cardio-vascular events, stroke or trauma).

The average values and standard deviations for each parameter in CBC pre-procedure and post-procedure, after the exclusion of the above-mentioned tests, was calculated for each patient. The percentage of the change (Δ) between pre- and post-procedure was calculated for each value as the fraction of pre-operative value. All the data were double checked and analyzed by two examiners.

### 2.3. Patient Stratificaiton and Data Analysis

Patients were categorized by age (under and over 55 years old), gender, suspected etiology of sialolithiases (DIS—drug induced sialolithiasis vs. OES—other etiologies of sialolithiasis) [4] and ASA (American Society of Anesthesiologists) score of general health [17]. To compare the findings between the groups, a 2-tailed paired Student’s t-test was performed utilizing IBM, SPSS Statistics for Windows, version 25.0 (IBM Ltd., Armonk, NY, USA). The significance of the difference between values is expressed as the *p*-value of Student’s t-test results (value of *p* < 0.05 was considered as significant).

### 2.4. Ethical Considerations

This study was carried out in accordance with the 1964 Declaration of Helsinki and was approved by the Institutional Review Board of Rabin Medical Center, Petach Tikva, Israel (Research number: RMC-18-0385).

## 3. Results

### 3.1. Study Group

Seventy-five patients who underwent a sialolith removal procedure at Rabin Medical Center, Petach Tikva, ISRAEL between 2013 and 2019, were included in this study. CBC results of pre- and post-sialolith removal interventions were gleaned from the patient files. Of the 75 patients 26 did not have available post-operative blood tests and were excluded from the study.

Forty-nine patients were included in the final analysis and their blood tests pre- and post-procedure were analyzed after application of exclusion criteria for confounding tests taken in acute clinical situations as described in the methodology section of this report.

From the 49 patients included in the study, 27 (55.1%) were male and 22 (44.9%) were female. Mean age of the patients was 53.06 (±17.37) years. The mean age of female patients was 50.71 (±17.36) years and the mean age of male patients was 54.91 (±17.41). 28 (57.1%) patients were found to be healthy and were classified as 1 according to the ASA Classification, 19 (38.7%)patients had controlled systemic diease (ASA = 2), and 2 (4.2%) of the patients had uncontrolled systemic disease (ASA = 3).

### 3.2. Comparison of Pre- and Post-Opeartive CBC for Entire Study Group

When examining the entire cohort (Table 1) CBC values show a significant post-operative increase in HCT (*p* < 0.01), HGB and RBC (*p* < 0.05). WBC shows a trend towards significant decrease (*p* = 0.079). No statistically significant differences were detected for MCH, MCHC, MCV and PLT.

### 3.3. Comparison of Pre- and Post-Opeartive CBC Stratified by Age

Stratification of the pre and post-operative CBC values by age (Table 2) with 55 years being the cut-off value, show a statistically significant (*p* < 0.05) increase in HCT and MCV in the >55-years old group. Furthermore, a trend for increase of the RBC is noted (*p* = 0.072). In age group <55-year-old, none of the parameters assessed in the CBC showed a significant trend.

### 3.4. Comparison of Pre- and Post-Opeartive CBC Stratified by Gender

Grouping by gender (Table 3) showed a statistically significant post-operative decrease in PLT in males (*p* < 0.05), and an increase in HCT (*p* < 0.05) and MCV (*p* = 0.05) in females. A trend for increase in post-operative RBC was noted in males (*p* = 0.079).

### 3.5. Comparison of Pre- and Post-Opeartive CBC Stratified by General Health Status

The effect of ASA classification is described in Table 4. The ASA = 1 group did not show any significant changes in the CBC pre- and post-operative values, while in the ASA ≥2 group, the HCT, HGB and RBC values were statistically significant increased post-operatively (*p* < 0.02).

### 3.6. Comparison of Pre- and Post-Opeartive CBC Stratified by Suspected Etiology of Sialolithiasis

Etiology of sialolithiasis was grouped by DIS and OES (Table 5). In the OES group, HGB showed a significant increase post-operatively (*p* < 0.02) and in the DIS group HCT was significantly increased post-operatively (*p* < 0.04).

## 4. Discussion

Sialolithiasis is the most common non-neoplastic disease of the salivary glands and the most common cause of salivary gland obstruction [1]. Sialolithiasis has been associated with a state of chronic inflammation and recurrent infections. Histopathological examination of sialolithiasis affected salivary glands demonstrates infiltration of the gland lobules by inflammatory cells which include mononuclear leukocytes [18]. Studies commonly reported a recovery of secretory function after sialolith removal [19].

Anemia of inflammation is a well-established pathology which is linked to immune mediation pathways. The physiological target of this immune response is aimed at depriving infectious pathogens of iron, creating a “functional iron deficiency”. This state is mostly mediated by the inflammatory cytokine IL-6, which interacts with iron regulatory hormone hepcidin and effects the ferrokinetic state by decrease of erythrocyte production and survival [9]. AI presents normochromic, normocytic anemia that is characteristically mild to moderate with hemoglobin levels of 7–12 g/dL. A prolonged state of inflammation may create a microcytic anemia, reflecting a true iron deficiency [20].

A leading pathogenetic pathway linking oral disease and systemic disease is the spread of bacteremia and/or cytokinemia from the oral cavity to the organs of the body [6]. The major cytokines reported as increasing in chronic periodontal infection and in chronic systemic diseases are TNF-alpha, IL-1, and IL-6 [21]. In the case of sialolithiasis, a chronic inflammation with exacerbations of bacterial infection of the afflicted gland occur [22]. As in any chronic inflammation, immune reactions are mediated by cytokine release, which in turn, have a systemic effect on all organs of the body, including the hematopoietic system.

In the periodontal literature, a link between chronic periodontal disease (infection) and a reduction in CBC parameters is well established. In a meta-analysis of 9 case control studies comprising 342 patients, França et al. [14] found a significant difference between periodontal patients and healthy controls. Average values of the periodontal patients were lower—HGB—11.8–13.67 g/dL vs. 12.8–16 g/dL; RBC—4.09–4.83 10^6^ µL, vs. 4.32–5.2 10^6^ µL. The conclusion was that periodontal disease leads to a decrease in CBC parameters.

In the present study, we aimed to investigate the association between sialolithiasis and surgical removal of the sialolith and AI. When analyzing the CBC results of the 49 patients included in the study, the average pre-operative values of HGB (13.55 ± 1.58 g/dL), HCT (41.41 ± 4.37%) and RBC (4.67 ± 0.48 10^6^ µL) were within normal range and did not indicate a diagnosis of anemia. However, when comparing these results to studies correlating periodontal disease to AI, the pre-operative values of sialolithiasis patients were within the range of decreased CBC parameters of the periodontal patients [14].

The effects of eradication of the underlying infectious/inflammatory disease by surgical treatment on blood parameters are well-documented in studies of periodontal management. In a study of 30 male periodontal patients with HGB values below 15 g/dL, Agarwal et al. [15] found that, 1 year after periodontal therapy, there was a significant increase in the HGB values (mean change +0.95 g/dL, *p* < 0.001) and the RBC values (mean change +0.22 10^6^ µL, *p* < 0.001). In an in-vivo study of experimentally induced peri-implantitis in dogs, Chaushu et al. showed that the induction of peri-implant disease caused a change in CBC values, which was corrected after surgical therapy [23]. Contrary to these findings, in a study of the effect of nonsurgical periodontal therapy on CBC parameters in 52 patients, whereas a clinical improvement in periodontal factors was noted, no significant changes in HGB or RBC counts were found [24].

In the present study group, a significant change between preoperative and postoperative parameters in the erythrocyte lineage of all patients included were found. Namely, an increased erythrocyte count, reflected in increased HCT and HGB, without a concomitant change in MCV or erythrocyte HGB content. These results are in accordance with improvement from a relative normocytic, normochromic anemic state, appropriate to the hallmark presentation of AI.

The increase in the RBC values is accompanied by a trend towards decrease in the white blood cell count and a not statistically significant decrease in the platelet count. The reduction in WBC and PLT in the complete blood count pre- and post-operatively may reflect evidence of reduction of hematopoietic markers in a mild chronic infectious process.

When categorizing the study group by gender, a significant change in blood count parameters was only noted in the hematocrit of women. Grouping the pre- and post-operative CBC values by age, with 55 years being the cutoff value, demonstrates a dissimilarity in improvement of CBC parameters between the examined age groups. While the younger age group did not demonstrate any significant improvement in blood parameters, the older age group shows a significant increase in hematocrit, mean corpuscular volume and a trend for an increase in the red blood cell count. In a similar manner, grouping by ASA score, found no significant improvement in the ASA = 1 group while in the ASA 2 and over group the HCT, HGB and RBC values were significantly increased post-operatively. Suspected etiology of sialolithiasis defined as drug-induced vs. other etiologies, showed a consistent trend for increase in blood parameters, but only the HCT in DIS group and the HGB in the OES group were statistically significant.

Patients in all age groups with specific diseases (ASA ≥ 2) and older patients are expected to experience an improvement in health-related quality of life from improvement in CBC parameters [7,9]. The fact that no change was demonstrated in the younger age group and in the ASA = 1 group may be attributed to a lack of inflammatory burden by concomitant chronic inflammatory diseases, and a more competent hematopoietic system that may hinder the manifestation of AI to an undetectable degree. Our results exhibit an improvement in CBC in the older age group and in the patients with co-morbidities. This may serve as an additional motivating factor when considering surgical intervention for the removal of sialoliths in the older age group and in patients with co-morbidities.

In a previous study on the same cohort of patients, we attempted to correlate different features of the patients and of the sialoliths with success in sialolith retrieval [25]. Age and ASA were examined as such predictors but were not found to serve as significant predictive variables. Therefore, even though these demographic features do not predict a positive surgical outcome, they may still predict a clinical benefit from surgical intervention.

The null-hypothesis of this study was that sialolithiasis and their surgical treatment are not associated with alterations in CBC. This hypothesis was rejected since we found a significant correlation between surgical treatment of sialolithiasis and an increase in HGB, HCT and RBC in the entire cohort. This current retrospective study may not observe the full magnitude of the true effect of sialolithiasis and sialolith removal on blood parameters. The heterogenicity in our cohort group and the retrospective study design may limit the methodological capacity to detect a suspected mild change. An additional collection of systemic inflammatory markers (C-Reactive Protein and Erythrocyte sedimentation rate) as well as iron and ferritin may provide further affirmation on the diagnosis of AI and the change in CBC parameters after surgical removal of the sialolith.

## 5. Conclusions

We found that sialolithiasis and sialolith removal surgery are associated with significant changes in CBC. Preoperative lower CBC values are corrected post-surgery. The speculated pathogenesis is relative anemia of inflammation. Further investigation on the effect of surgical treatment of chronic inflammatory conditions on the complete blood count values is warranted.

## Figures and Tables

**Table 1 biology-10-00414-t001:** Change between pre-operative and post-operative CBC parameters (*n* = 49).

	PLT (10^3^ μL)	MCV (fL)	MCHC (gr/dL)	MCH (pg)	HCT (%)	HGB g/dL)	RBC (10^6^ μL)	WBC (10^3^ μL)
Pre-operative average	233.83 ± 51.98	89.04 ± 6.98	32.78 ± 1.20	29.01 ± 2.28	41.41 ± 4.37	13.55 ± 1.58	4.67 ± 0.48	6.95 ± 2.05
Post-operative average	228.43 ± 54.39	89.42 ± 7.11	32.58 ± 1.17	29.23 ± 2.67	42.25 ± 4.33	13.72 ± 1.60	4.75 ± 0.53	6.71 ± 1.87
Δ	−5.40 ± 27.94	0.38 ± 5.32	−0.20 ± 1.13	0.23 ± 1.41	0.84 ± 2.18	0.18 ± 0.63	0.07 ± 0.23	−0.23 ± 0.92
Average of Δ as % of pre-operative result	−2.31	0.43	−0.62	0.78	2.03	1.32	1.57	−3.38
Significance (*t*-test *p* value)	0.182	0.62	0.212	0.267	0.007	0.05	0.033	0.079

**Table 2 biology-10-00414-t002:** Change between pre-operative and post-operative CBC parameters stratified by age.

**Age < 55 years (*n* = 25)**	**PLT (10^3^ μL)**	**MCV (fL)**	**MCHC (gr/dL)**	**MCH (pg)**	**HCT (%)**	**HGB (g/dL)**	**RBC (10^6^ μL)**	**WBC (10^3^ μL)**
Pre-operative average	241.37 ± 43.24	89.61 ± 7.37	32.99 ± 1.34	29.03 ± 2.69	41.26 ± 4.88	13.54 ± 1.76	4.68 ± 0.53	7.25 ± 2.21
Post-operative average	235.08 ± 47.01	88.24 ± 8.00	32.76 ± 1.39	28.94 ± 3.18	41.67 ± 4.33	13.67 ± 1.75	4.75 ± 0.50	6.91 ± 2.12
Δ	−6.30 ± 22.54	−1.37 ± 5.66	−0.24 ± 1.40	−0.09 ± 1.02	0.40 ± 2.13	0.13 ± 0.58	0.07 ± 0.26	−0.34 ± 1.01
Average of Δ as % of pre-operative result	−2.61	−1.53	−0.72	−0.31	0.98	0.94	1.45	−4.68
Significance (*t*-test *p* value)	0.175	0.239	0.405	0.66	0.354	0.287	0.208	0.105
**Age > 55 years (*n* = 24)**	**PLT (10^3^ μL)**	**MCV (fL)**	**MCHC (gr/dL)**	**MCH (pg)**	**HCT (%)**	**HGB (g/dL)**	**RBC (10^6^ μL)**	**WBC (10^3^ μL)**
Pre-operative average	225.98± 59.68	88.46 ± 6.67	32.56 ± 1.01	28.99 ± 1.80	41.56 ± 3.87	13.55 ± 1.40	4.67 ± 0.45	6.63 ± 1.86
Post-operative average	221.50± 61.40	90.66 ± 5.96	32.39 ± 0.88	29.54 ± 2.05	42.86 ± 4.34	13.78 ± 1.47	4.75 ± 0.57	6.50 ± 1.58
Δ	−4.47± 33.13	2.20 ± 4.35	−0.17 ± 0.78	0.56 ± 1.69	1.30 ± 2.18	0.23 ± 0.68	0.08 ± 0.20	−0.13 ± 0.81
Average of Δ as % of pre-operative result	−1.98	2.49	−0.52	1.92	3.13	1.71	1.69	−1.89
Significance (*t*-test *p* value)	0.515	0.021	0.299	0.12	0.008	0.111	0.072	0.458

**Table 3 biology-10-00414-t003:** Change between pre-operative and post-operative CBC parameters stratified by gender.

**Male (*n* = 27)**	**PLT (10^3^ μL)**	**MCV (fL)**	**MCHC (gr/dL)**	**MCH (pg)**	**HCT (%)**	**HGB (g/dL)**	**RBC (10^6^ μL)**	**WBC (10^3^ μL)**
Pre-operative average	233.28 ± 51.99	90.95 ± 5.59	33.01 ± 1.34	29.49 ± 1.90	43.96 ± 3.25	14.44 ± 1.27	4.91 ± 0.45	7.50 ± 2.14
Post-operative average	221.62 ± 49.47	89.97 ± 5.49	32.88 ± 1.15	29.74 ± 2.08	44.62 ± 3.36	14.58 ± 1.35	4.98 ± 0.48	7.23 ± 1.89
Δ	−11.66 ± 29.23	−0.98 ± 5.50	−0.13 ± 1.21	0.25 ± 1.26	0.66± 2.24	0.15 ± 0.60	0.07 ± 0.21	−0.26 ± 0.96
Average of Δ as % of pre-operative result	−5.00	−1.07	−0.38	0.86	1.51	1.01	1.49	−3.51
Significance	0.048	0.364	0.592	0.307	0.135	0.216	0.079	0.167
**Female (*n* = 22)**	**PLT (10 ^3^ μL)**	**MCV (fL)**	**MCHC (gr/dL)**	**MCH (pg)**	**HCT (%)**	**HGB (g/dL)**	**RBC (10^6^ μL)**	**WBC (10^3^ μL)**
Pre-operative average	234.51 ± 53.18	86.71 ± 7.90	32.51 ± 0.97	28.42 ± 2.59	38.28 ± 3.47	12.45 ± 1.19	4.39 ± 0.37	6.27 ± 1.73
Post-operative average	236.78 ± 60.00	88.75 ± 8.80	32.21 ± 1.11	28.61 ± 3.20	39.34 ± 3.59	12.67 ± 1.22	4.46 ± 0.44	6.07 ± 1.67
Δ	2.27 ± 24.80	2.04 ± 4.70	−0.30 ± 1.04	0.19 ± 1.60	1.06 ± 2.14	0.22 ± 0.68	0.07 ± 0.27	−0.20 ± 0.88
Average of Δ as % of pre-operative result	0.97	2.36	−0.92	0.68	2.77	1.74	1.68	−3.18
Significance (*t*-test *p* value)	0.672	0.05	0.19	0.576	0.031	0.147	0.21	0.298

**Table 4 biology-10-00414-t004:** Change between pre-operative and post-operative CBC stratified by ASA status.

**ASA = 1 (*n* = 28)**	**PLT (10^3^ μL)**	**MCV (fL)**	**MCHC (gr/dL)**	**MCH (pg)**	**HCT (%)**	**HGB (g/dL)**	**RBC (10^6^ μL)**	**WBC (10^3^ μL)**
Pre-operative average	233.07 ± 44.09	90.72 ± 4.91	32.88 ± 1.27	29.33 ± 1.75	41.73 ± 3.54	13.65 ± 1.34	4.67 ± 0.48	6.67 ±. 1.99
Post-operative average	228.61 ± 45.19	90.08 ± 5.42	32.59 ± 1.15	29.33 ± 1.97	41.87 ± 3.34	13.67 ± 0.877	4.67 ± 0.46	6.5 s2± 1.87
Δ	−4.46 ± 2.71	−0.64 ± 5.56	−0.29 ± 1.28	0.01 ± 0.90	0.15 ± 1.87	0.02 ± 0.57	0.00 ± 0.17	−0.16 ± 0.96
Average of Δ as % of pre-operative result	−1.91	−0.71	−0.89	0.02	0.35	0.12	−0.05	−2.35
Significance (*t*-test *p* value)	0.264	0.547	0.235	0.967	0.687	0.877	0.947	0.396
**ASA ≥ 2 (*n* = 21)**	**PLT (10^3^ μL)**	**MCV (fL)**	**MCHC (gr/dL)**	**MCH (pg)**	**HCT (%)**	**HGB (g/dL)**	**RBC (10^6^ μL)**	**WBC (10^3^ μL)**
Pre-operative average	234.85 ± 62.11	86.82 ± 8.68	32.65 ± 1.11	28.58 ± 2.82	40.98 ± 5.36	13.41 ± 1.87	4.68 ± 0.5	7.31 ± 2.11
Post-operative average	228.19 ± 65.99	88.56 ± 8.95	32.57 ± 1.21	29.10 ± 3.34	42.75 ± 1.45	13.80 ± 0.46	4.85 ± 0.6	6.97 ± 1.88
Δ	−6.66 ± 35.95	1.74 ± 4.79	−0.09 ± 0.91	0.52 ± 1.88	1.77 ± 2.24	0.39 ± 0.66	0.17 ± 0.27	−0.34 ± 0.86
Average of Δ as % of pre-operative result	−2.84	2	−0.26	1.81	4.32	2.94	3.72	−4.63
Significance (*t*-test *p* value)	0.406	0.112	0.671	0.22	0.002	0.012	0.008	0.087

**Table 5 biology-10-00414-t005:** Change between pre-operative and post-operative CBC stratified by suspected etiology.

**OES (*n* = 13)**	**PLT (10^3^ μL)**	**MCV (fL)**	**MCHC (gr/dL)**	**MCH (pg)**	**HCT (%)**	**HGB (g/dL)**	**RBC (10^6^ μL)**	**WBC (10^3^ μL)**
Pre-operative average	239.58 ± 41.44	92.01 ± 5.91	33.18 ± 1.67	29.48 ± 2.41	41.95 ± 4.53	13.78 ± 1.71	4.69 ± 0.57	6.57 ± 2.21
Post-operative average	242.85 ± 50.63	90.57 ± 6.02	32.87 ± 1.38	29.74 ± 2.18	42.80 ± 4.05	14.10 ± 1.71	4.75 ± 0.50	6.46 ± 2.43
Δ	3.26 ± 21.28	−1.44 ± 7.69	−0.31 ± 1.69	0.26± 0.67	0.85 ± 1.80	0.32 ± 0.40	0.06 ± 0.17	−0.12 ± 0.03
Average of Δ as % of pre-operative result	1.36	−1.57	−0.94	0.87	2.03	2.31	1.2	−1.76
Significance (*t*-test *p* value)	0.591	0.512	0.518	0.197	0.115	0.014	0.249	0.695
**DIS (*n* = 36)**	**PLT (10^3^ μL)**	**MCV (fL)**	**MCHC (gr/dL)**	**MCH (pg)**	**HCT (%)**	**HGB (g/dL)**	**RBC (10^6^ μL)**	**WBC (10^3^ μL)**
Pre-operative average	231.76 ± 55.68	87.97 ± 7.11	32.64 ± 0.97	28.84 ± 2.24	41.21 ± 4.37	13.46 ± 1.54	4.67 ± 0.46	7.08 ± 2.00
Post-operative average	223.22 ± 55.44	89.01 ± 7.50	32.47 ± 1.08	29.05 ± 2.84	42.05 ± 4.47	13.59 ± 1.56	4.75 ± 0.55	6.80 ± 1.65
Δ	−8.53 ± 29.62	1.04 ± 4.11	−0.17 ± 0.87	0.22 ± 1.60	0.84 ± 2.33	0.13 ± 0.69	0.08 ± 0.26	−0.28 ± 0.88
Average of Δ as % of pre-operative result	−3.68	1.18	−0.51	0.75	2.04	0.95	1.7	−3.92
Significance (*t*-test *p* value)	0.093	0.14	0.264	0.425	0.038	0.277	0.07	0.067

DIS—Drug Induced Sialolithiasis, OES—Other Etiologies of Sialolithiasis.

## Data Availability

All data generated in this study are shown in this article and its tables.

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
