# Peer review of "Surgical Treatment of Sialolithiasis Leads to Improvement in the Complete Blood Count"

_biology, 2021, doi:10.3390/biology10050414_

Round 1

Reviewer 1 Report

The manuscript submitted to Biology entitled “Surgical Treatment of Sialolithiasis Leads to Improvement in the Complete Blood Count” is an original article which aim to find whether removal of the sialolith and alleviation of the inflammation affect the complete blood count results speculating about anemia of inflammation.

On my opinion the article is well written, with good English. The authors' hypothesis is interesting.

I highlighted some issues:

  • Minor spell check required
  • Summary of abbreviations required
  • Abstract: Please structure the abstract to attract the reader's attention, and to adapt it accordingly.
  • Introduction: The section has been adequately prepared, considering the authors' hypothesis.
  • Materials and methods: Anemia due to known causes (aplastic anemia, sickle cell anemia, rheumatoid arthritis, leukemia, iron deficiency anemia, vitamin deficiencies) is not mentioned among the exclusion criteria. Please improve.
  • Results: Reading this section I could not understand exactly how many patients were included, their gender, their pathologies, etc. Please improve.
  • Discussion: The section has been adequately prepared, considering the authors' hypothesis.
  • Conclusions: Further studies will be needed to confirm the authors' hypotheses.

After making the indicated changes. I am available for a second round of review.

Author Response

Dear Reviewer

Thank you for your diligent review of our article. Please find below our detailed response to your remarks. The changes made were highlighted in the text using “track changes” in Word.

  • On my opinion the article is well written, with good English. The authors' hypothesis is interesting.
    • Thank you
  • Minor spell check required
    • Found and corrected typos, see in "tracked changes" in word document.
  • Summary of abbreviations required
    • According to MDPI instructions to authors: "Abbreviations should be defined in parentheses the first time they appear in the abstract, main text, and in figure or table captions and used consistently thereafter".
    • We removed abbreviations from the abstract and confirmed correct styling throughout the ms
  • Abstract: Please structure the abstract to attract the reader's attention, and to adapt it accordingly.
    • Revised the simple summary and abstract
  • Introduction: The section has been adequately prepared, considering the authors' hypothesis.
    • Thank you
  • Materials and methods: Anemia due to known causes (aplastic anemia, sickle cell anemia, rheumatoid arthritis, leukemia, iron deficiency anemia, vitamin deficiencies) is not mentioned among the exclusion criteria. Please improve.
    • Thank you for this important remark. After careful re-inspection of the patient files, fortunately, none of the patients included had a diagnosis of anemia due to a known cause.
    • Added anemia of a known cause to exclusion criteria.
  • Results: Reading this section I could not understand exactly how many patients were included, their gender, their pathologies, etc. Please improve.
    • Amended – added section 3.1 to results – describing the patient group and demographics.
  • Discussion: The section has been adequately prepared, considering the authors' hypothesis.
    • Thank you
  • Conclusions: Further studies will be needed to confirm the authors' hypotheses.
    • Added to conclusions

Reviewer 2 Report

The subject of the article is interesting; however, the introduction section may be highlighted the relationship between oral chronic diseases and systemic diseases. also, in the discussion section, it could be underlined the better bi-directional link between the two diseases and the role of chronic inflammation in this link

I suggest that the manuscript be evaluated by a statistician and leave the decision to the Editor's discretion.

Author Response

Dear Reviewer

Thank you for your diligent review of our article. Please find below our detailed response to your remarks. The changes made were highlighted in the text using “track changes” in Word.

  • The subject of the article is interesting; 
    • Thank you

  • however, the introduction section may be highlighted the relationship between oral chronic diseases and systemic diseases.
    • Added paragraph (4th paragraph) to introduction highlighting this relationship as per your suggestion.

  • also, in the discussion section, it could be underlined the better bi-directional link between the two diseases and the role of chronic inflammation in this link-
  • Added paragraph to discussion detailing the propositioned link between sialolithiases and AI

Reviewer 3 Report

This paper is the study about “Surgical Treatment of Sialolithiasis Leads to Improvement in the Complete Blood Count”. We think that this study is interesting. However, it has a small number of patients and is not suitable for this high quality paper. Please make the purpose of this study clear and consider reconsideration.  Some comments are as below:

  1. You should not be used an abbreviation in Abstract.
  2. It has a small number of patients and is not suitable for this high quality paper.
    There are 342 patients with chronic periodontal disease, but only 49 in this paper.
  3. Please elaborate on its relevance to other diseases.
  4. Please add references. There are very few references and the discussion is unclear.

Author Response

Dear Reviewer 3

Thank you for your diligent review of our article. Please find below our detailed response to your remarks. The changes made were highlighted in the text using “track changes” in Word.

  • You should not be used an abbreviation in Abstract.
  • Corrected

  • It has a small number of patients and is not suitable for this high-quality paper.
    There are 342 patients with chronic periodontal disease, but only 49 in this paper.
  • The prevalence of periodontal disease in the adult population is up to 50%. As stated in our introduction, the prevalence of sialoliths is around 1%, a significant part of these cases are a-symptomatic and most of them do not need to undergo surgery in general anesthesia to remove the stone. We believe that statistically significant results analyzed from a cohort of 49 patients who have retrievable relevant blood counts pre- and post-surgery is a robust enough set of data to be published. Furthermore, the statistical analysis uses t-tests without sophisticated statistical manipulations.

  • Please elaborate on its relevance to other diseases.
  • Thank you for this comment. According this suggestion a paragraph demonstrating the distinctive examples of bi-lateral link between oral chronic diseases and systemic diseases was added to the introduction

  • Please add references. There are very few references and the discussion is unclear.
  • Following this suggestion and additional comments by other reviewers changes were made to the article and additional references were added.

Round 2

Reviewer 1 Report

After the changes made the article may be suitable for publication.